# The Effects of Using Cements of Different Thicknesses and Amalgam Restorations with Different Young's Modulus Values on Stress on Dental Tissue: An Investigation Using Finite Element Analysis

Hakan Yasin Gönder [1,*], Mehmet Gökberkkaan Demirel [2], Reza Mohammadi [3], Sinem Alkurt [1], Yasemin Derya Fidancioğlu [4] and Ibrahim Burak Yüksel [5]

1 Department of Restorative Dentistry, Faculty of Dentistry, Necmettin Erbakan University, Konya 42090, Turkey
2 Department of Prosthetic Dentistry, Faculty of Dentistry, Necmettin Erbakan University, Konya 42090, Turkey
3 Faculty of Dentistry, Necmettin Erbakan University, Konya 42090, Turkey
4 Department of Pediatric Dentistry, Faculty of Dentistry, Necmettin Erbakan University, Konya 42090, Turkey
5 Department of Oral Diagnose and Radiology, Faculty of Dentistry, Necmettin Erbakan University, Konya 42090, Turkey
* Correspondence: hygonder@erbakan.edu.tr; Tel.: +90-5305815589; Fax: 03322200045

**Abstract: Background:** In this study, it was aimed to use a finite element stress analysis method to determine the amount of stress on enamel, dentin, restoration, resin cement and glass ionomer cement in amalgam class II disto-occlusal (DO) cavities by using two different cements with different thicknesses and amalgams with different Young's modulus values, respectively. **Methods:** A three-dimensional tooth model was obtained by scanning an extracted human maxillary first molar with dental tomography. A class II DO cavity including 95-degree cavity margin angles was created. Resin cement (RC) and glass ionomer (GI) cement with different Young's modulus measures (RC: 7.7 GPa, GI: 10.8 GPa) were used in amalgam. Different thickness combination groups were simulated: 50 μm, 100 μm and 150 μm. Additionally, amalgams with different Young's modulus values were used with the same thickness of different cements (Amalgam Young's modulus: 35 GPa and 50 GPa). A load of 600 N was delivered to the chewing area. The stress distributions on enamel, dentin, restoration, resin cement and class ionomer cement were then analyzed using finite element analysis. **Results:** The most stress accumulation was observed in the enamel tissue across all groups where resin cement or glass ionomer cement were used in different thicknesses and where amalgam restorations were used with different Young's modulus values. The least stress accumulation was observed in the cement itself. **Conclusions:** According to the results obtained, there was no difference between the two cement types in terms of stress accumulations in the models. However, when the same cements with different thicknesses were evaluated, it was concluded that the presence of both glass ionomer and resin cement with a thickness of 150 μm causes less stress on the restoration surface. Furthermore, when the cements were combined with different thicknesses and with different amalgam Young's modulus values, it was concluded that 50 GPa causes less stress on restoration surface.

**Keywords:** amalgam; finite element analysis; resin cement; glass ionomer cement; stress distribution

## 1. Background

The chewing motion creates significant reactive stresses in teeth and supporting tissues in different directions and sizes. It is known that dental hard and support tissues are a complex combination with various mechanical properties [1,2].

Due to the complexity of tooth structure and diversity of restorative materials, the mechanical performance of a restored tooth has been poorly understood so far. Although the mechanical benefits of curved junctions were reported by researchers, such junctions

have not been used in dental restoration and cavity design. Nevertheless, methods to analyze 3D models of restoration and teeth, including the influence of the modulus of elasticity, Poisson's ratio, or shape optimization of restoration, have shown favorable results in strengthening tooth restoration structure in previous studies [3–5].

Finite element analysis is a numerical method that has been used in solving many problems in physics and engineering since the early 1900s. It is one of the most effective methods that can also be used in stress, strength, fluid, vibration and dynamic calculations. The finite element analysis method can be defined as a solution method in which complex problems are divided into simple sub-problems and each one is solved separately [6,7]. Based on this definition, the principle of the method is that after each object is divided into a certain number of elements, they are in contact with the points formed at the corners where these elements intersect with each other. Finite element stress analysis allows for more successful restorations and is frequently used in dentistry research, as the stress and strains occurring in all materials, including living tissues, can be calculated [8]. A finite element model of a 3D scanned tooth subjected to simulated masticatory forces is used to carry out stress distribution analysis in various conditions for a premolar tooth. Finite element models that are easily modifiable and able to provide quick results become more and more prominent with the advances in finite element software capabilities [9].

An appropriate restorative material that can fully replace that of human teeth with all their biological and mechanical properties has not yet been found. Human teeth have more complex conformation and much better properties and biocompability than all available dental restorative materials [10]. Restorative materials should be selected to correct and replace the lost tooth structure. Although the ideal material for dental tissue is not yet available, the restorative material should be able to replace both enamel and dentin, and the elastic properties of materials should be similar to dental tissues [11].

Amalgam has been a widely used material for many years; it is not the only filling material, but it has advantages over other materials. It is easier to place than composites or other tooth-colored filling materials because it does not require a perfectly isolated working area. This can be challenging for people with special needs or dentists who work in very crowded hospitals, such as oral and dental health centers in Turkey. Thus, it is still frequently used in these centers because of its ease of use. It is economical and has long-term clinical performance. However, its retention, which is only mechanical, limits its use when minimal cavity preparation is required; a significant amount of healthy dental tissue must be removed to provide adequate resistance and retention in amalgam cavities. It has also disadvantages, such as poor aesthetic qualities, corrosion and sometimes allergy. It has mercury toxicity, which is a major concern. Most of the forces generated during chewing movements are on the occlusal surface and edges of the restorations. Therefore, it is seen as a disadvantage in terms of fracture strength for amalgams with low tensile strength, and this results in marginal breakdown in the long term [12–14].

With the use of the base material, it is aimed both to protect the pulp tissue and to control the stresses caused by occlusal forces. Glass ionomer cements have been used in many clinical applications since 1970 due to their biocompatibility and fluoride ion release [15]. Glass ionomer cements are materials that have the capacity to cross-link with calcium ions in the tooth. Thus, it can be bonded to both dental tissues and metals with direct adhesion. Low-wear resistance, short working time and fragile structure are the factors limiting the usage area of the plinth material [16]. Resin cements have been developed to improve the negative aspects of conventional cements such as high solubility, lack of adhesion and inadequate aesthetic properties.

The aim of this study was to use a finite element stress analysis method to determine the amount of stress on enamel, dentin, restoration, resin cement and glass ionomer cement in amalgam class 2 disto-occlusal (DO) cavities by using cements with different thicknesses and amalgams with different Young's modulus values.

## 2. Materials and Methods

For the first step, a 3D image of an extracted left permanent maxillary first molar tooth was scanned with a dental tomography device (J Morita MFG Corp., Kyoto, Japan). The size of the imaging volume was a cyclinder with diameter 40 × height 40 mm at the X-ray rotational center. Images were taken under the exposure condition of 90 kVp (X-ray tube voltage) and 5 mA (value of electric current), which were the standard parameters and can be changed for different subjects. Images were taken using 160 qm and 17.5 s exposure time parameters. The 3D image created using Geomagic Design X 2020.0 software was divided into surfaces and the necessary arrangements were made. The periodontal ligament (PDL) was not designed, so fixed and pinned boundary conditioning was used to simulate roots fixed in the bone. The tooth model was placed in the coordinate system so that the *x*-axis defined the buccolingual direction, the *y*-axis defined the mesiodistal direction and the *z*-axis was oriented upwards (Figure 1). Using Solidworks 2013 software (Solidworks Corp., USA), class 2 DO cavity modeling was performed on the 3D model with a cavity angle of 95 degrees (Figure 2).

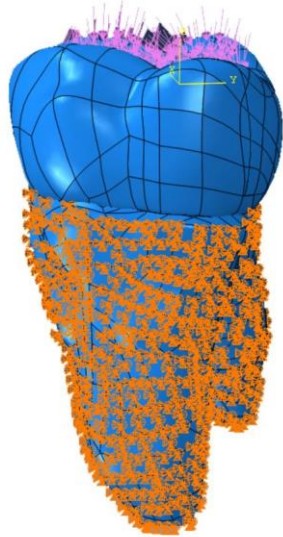

**Figure 1.** Load and boundary conditions.

As a potential simplification for the complexity of finite element analysis (FEA) simulations, the assumption was made of linear elastic isotropic material behavior; it was assumed that the material properties of enamel, dentin, resin cement and filling material are isotropic and linearly elastic (Table 1).

**Table 1.** Mechanical properties of tissues and materials used in 3D finite element stress analysis models of maxillary molars.

| Material | Young's Modulus (GPa) | Poisson's Ratio | Tensile Strength (MPa) | Compressive Strength (MPa) |
|---|---|---|---|---|
| Dentin | 18.6 [17] | 0.31 [17] | 98.7 [18] | 297.0 [18] |
| Enamel | 84.1 [17] | 0.33 [17] | 10.3 [18] | 384.0 [18] |
| Amalgam | 35.0 [19] 50.0 [19] | 0.35 [19] 0.29 [19] | 3–58 [20] | 45–550 [20] |
| Resin Cement | 7.7 [21] | 0.3 [21] | 98 [21] | 262 [21] |
| Glass Ionomer | 10.8 [22] | 0.3 [22] | - | - |

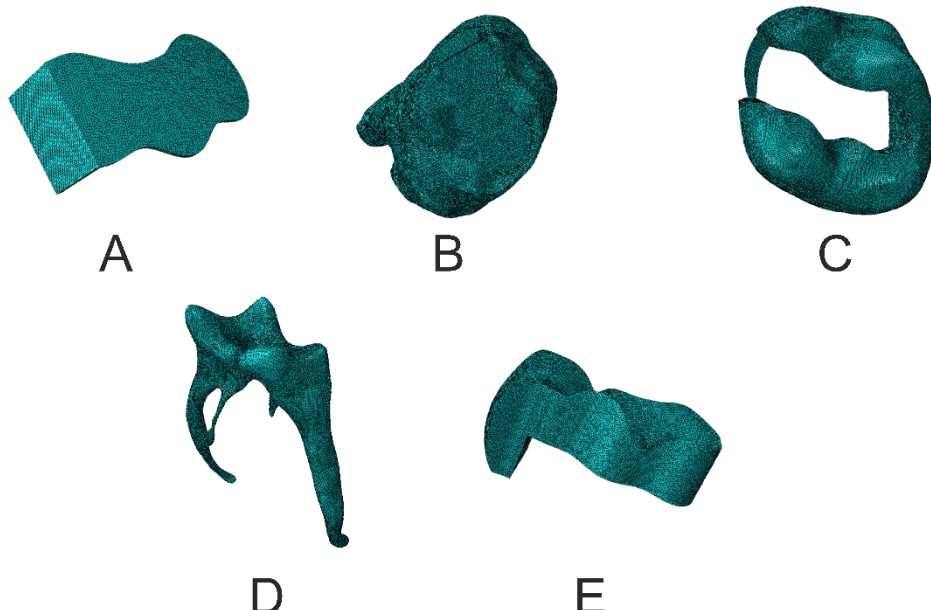

**Figure 2.** Three-dimensional tooth model structures ((**A**): cement, (**B**): dentin, (**C**): enamel, (**D**): pulp, (**E**): restoration).

A cavity was created in a computer model. A class II cavity with an occlusal depth of 4 mm and an occlusal-gingival depth of 6 mm was fixed with the occlusal margin in the enamel and the gingival margin in the dentin. The interior line angles of the cavity were rounded (smoothed) to avoid any stress concentration. Different combinations were then simulated: two cement groups (RC and GI) with the same modulus of Poisson's ratio and different Young's modulus values and different thicknesses (50 μm: A, 100 μm: B and 150 μm: C) were used. Therefore, six study groups were created (Table 2).

**Table 2.** Study groups.

| Study Group | Cement Thickness |
|:---:|:---:|
| RC | 50 μm |
| RC | 100 μm |
| RC | 150 μm |
| GI | 50 μm |
| GI | 100 μm |
| GI | 150 μm |

RC: Resin Cement; Young's modulus (GPa): 7.7. GI: Glass Ionomer Cement; Young's modulus (GPa): 10.8.

The materials used were included in a simulation of isotropic linear elastic restoration. A load of 600 N was delivered to the chewing area. This force was delivered to both restoration and tooth surface. The stress distribution was analyzed with FEA using the Abaqus software program (2020 Dassault Systems Simulation Corp., Johnson, RI, USA). The mesh, nodes, and element properties used in FEA for the tooth and cement thickness are shown in Table 3.

**Table 3.** Nodes and elements for the different tested restorations.

| Model | Total Elements | Total Nodes | Mesh Type |
|:---:|:---:|:---:|:---:|
| 50 μ | 7,428,602 | 1,347,225 | Linear tetrahedral elements of C3D4 |
| 100 μ | 7,445,941 | 1,350,049 | Linear tetrahedral elements of C3D4 |
| 150 μ | 7,457,979 | 1,352,224 | Linear tetrahedral elements of C3D4 |

## 3. Results

According to the findings, the greatest stress accumulation was observed in enamel tissue in all groups where either resin or glass ionomer cement were used in different thickness or amalgam was used in different Young's modulus. The least stress accumulation was observed in cement material itself.

Firstly, in the groups where amalgam placed in 35 GPa, we observed enamel after resin cement was applied; the greatest stress accumulation was found in the presence of 150 μm thick cement, and it was found to be 90.25 MPa. Stress accumulation decreased when cement thickness also decreased. It was found to be 89.62 MPa in the presence of 100 μ-thick cement; on the other hand, it was found to be 89.12 MPa in 50 μm thick cement (Figure 3).

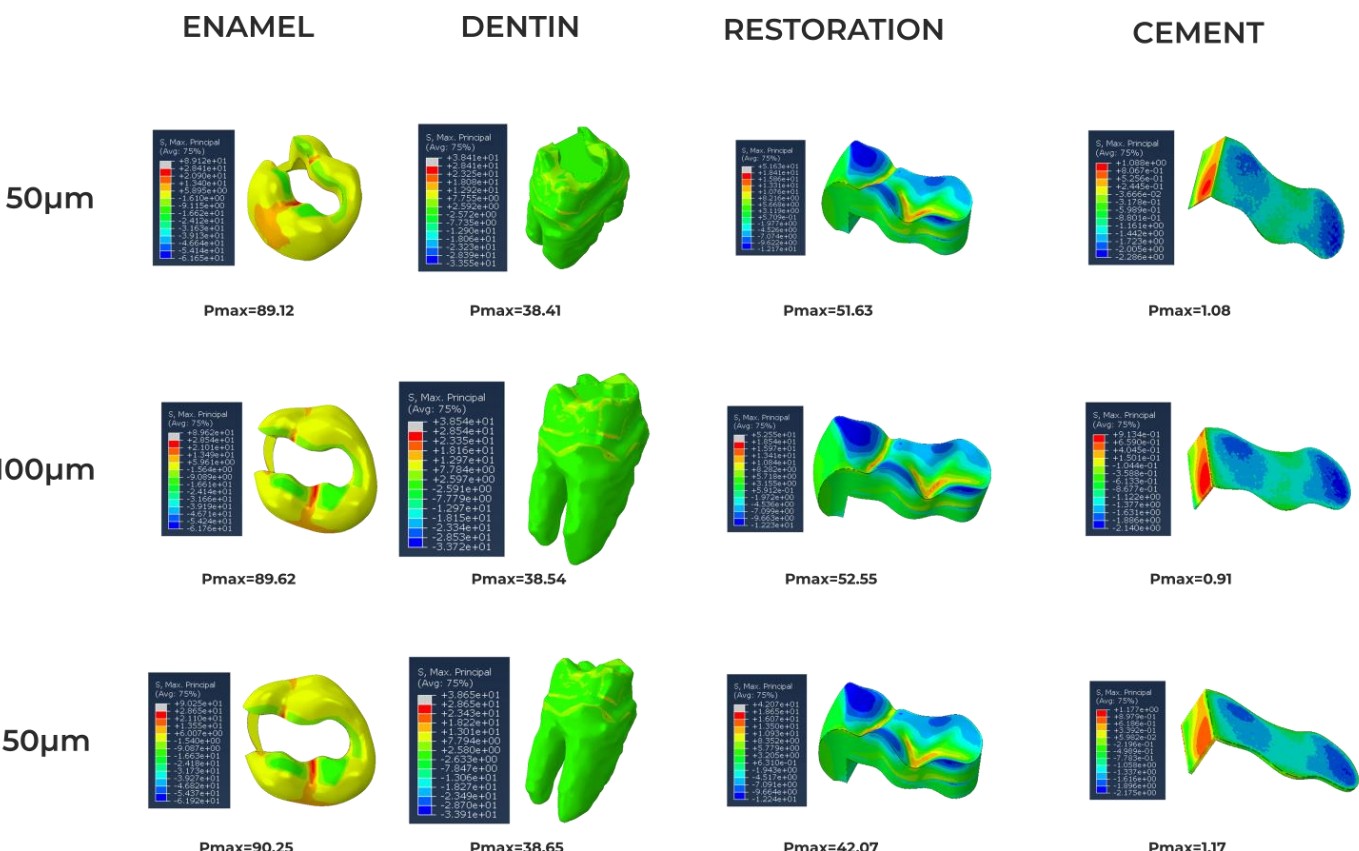

**Figure 3.** Maximum principal stress distribution in the enamel, dentin and restoration, compared with different thicknesses of RC used with the amalgam (35 GPa Young's modulus of amalgam).

It was observed that dentin tissue was not different to enamel tissue in terms of cement thickness and stress decreasing. 38.65 MPa stress accumulation was observed in the presence of 150 μm thick cement, 38.54 MPa in 100 μm thick cement and 38.41 MPa in 50 μm thick cement (Figure 4).

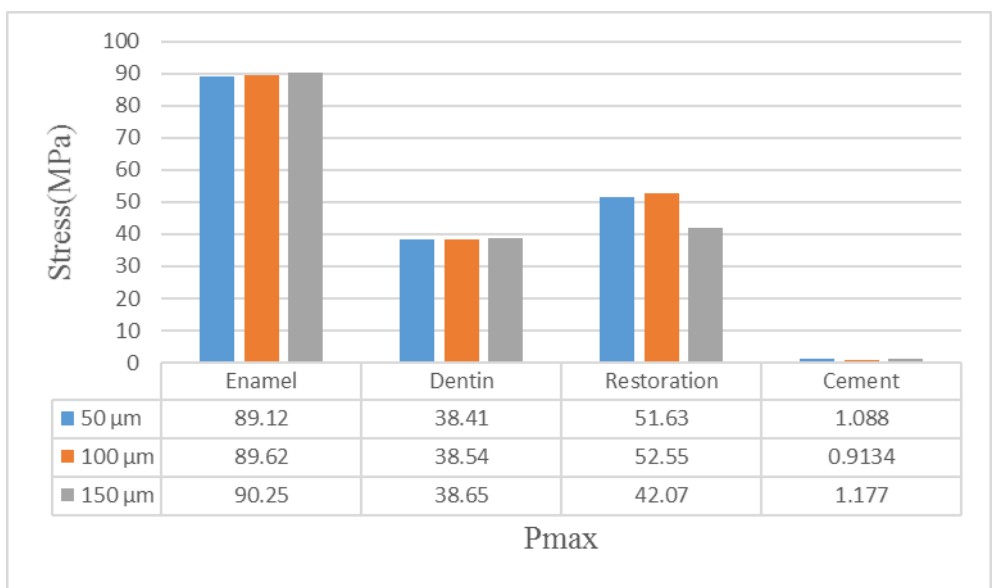

**Figure 4.** Maximum principal stress distribution in restorations with RC in different thicknesses (35 GPa Young's modulus of amalgam).

When we observed stress accumulations on the restoration surface, it was found to be 42.07 MPa in the presence of 150 μm-thick cement, 52.55 MPa in 100 μm thick cement and 51.63 MPa in 50 μm thick cement (Figure 4).

In all resin cement-applied groups, the least stress accumulation was found in the cement material itself. Stress accumulation was found to be 1.177 MPa in the presence of 150 μm thick cement, 0.9134 MPa in 100 μm thick cement and 1.088 MPa in 50 μm thick cement (Figure 4).

When the stress accumulations in the enamel tissue were compared in the groups with glass ionomer cement applied, 89.99 MPa was observed in the presence of 150-μm thick cement, 89.47 MPa in the presence of 100 μm thick cement and 89.06 MPa in the presence of 50 μm thick cement (Figure 5).

It was observed that in dentin tissue after glass ionomer cement was applied, 38.5 MPa stress accumulation was found in the presence of 150 μm thick cement, 38.43 MPa in the presence of 100 μm thick cement and 38.36 MPa in the presence of 50 μm thick cement (Figure 6).

After comparison of stress accumulation on the restoration surface in the groups with glass ionomer applied, 42.28 MPa was observed in the presence of 150 μm thick cement, 52.75 MPa in the presence of 100 μm thick cement and 51.73 MPa in 50 μm thick cement presence (Figure 6).

In all glass ionomer applied groups, the least stress accumulation was found in cement material itself. 1.686 MPa was observed in the presence of 150 μm thick cement, 1.295 MPa in the presence of 100 μm thick cement and 1.277 MPa in the presence of 50 μm thick cement (Figure 6).

In the groups where amalgam placed in 50 GPa, we observed enamel after resin cement was applied and the greatest stress accumulation was found in the presence of 150 μm thick cement, and it was found to be 88.14 MPa. Stress accumulation decreased when cement thickness also decreased. It was found to be 87.45 MPa in the presence of 100 μm thick cement; on the other hand, it was found to be 86.89 MPa in 50 μm thick cement (Figure 7).

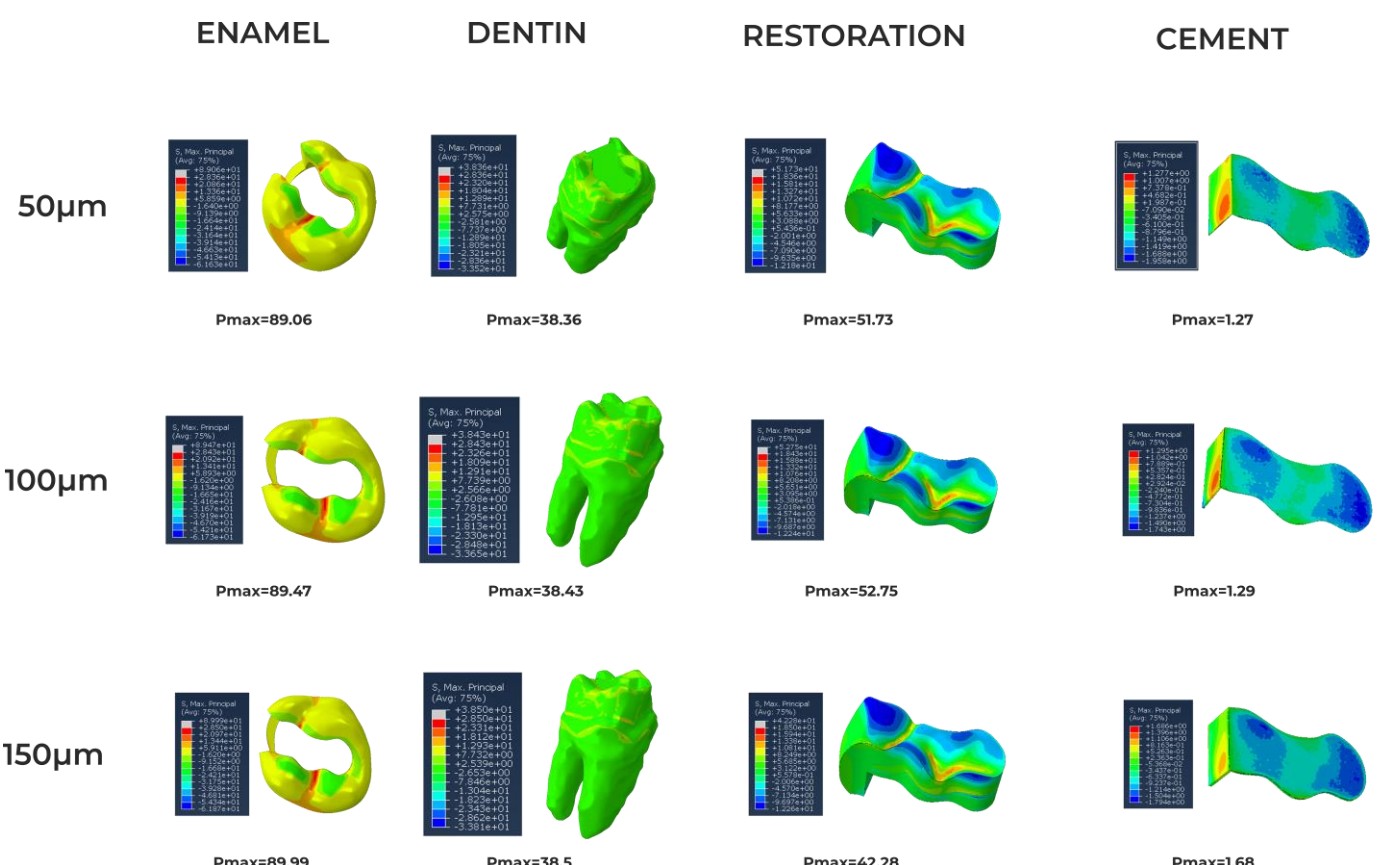

**Figure 5.** Maximum principal stress distribution in the enamel, dentin and restoration with different thicknesses of GI cement used with the amalgam (35 GPa Young's modulus of amalgam).

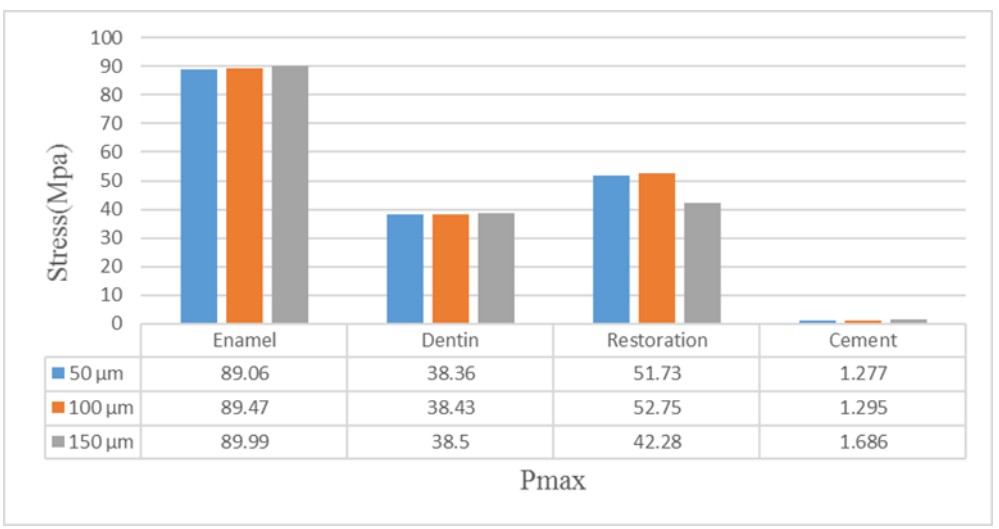

**Figure 6.** Maximum principal stress distribution in restorations applied with GI cements of different thicknesses (35 GPa Young's modulus of amalgam).

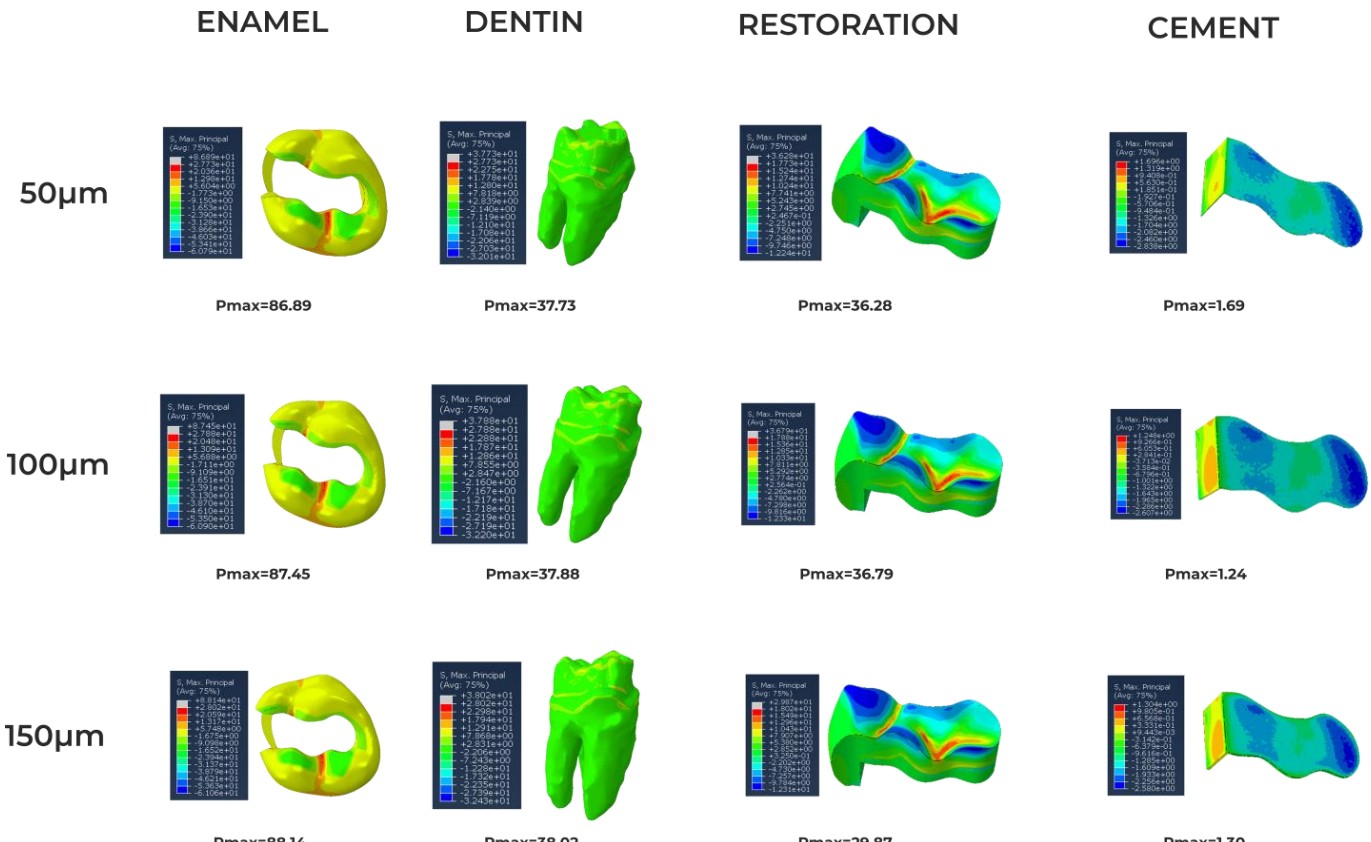

**Figure 7.** Maximum principal stress distribution in the enamel, dentin and restoration with different thicknesses of RC used with the amalgam (50 GPa Young's modulus of amalgam).

It was observed that dentin tissue was not different than enamel tissue in the cement thickness and stress decreasing. A 38.02 MPa stress accumulation was observed in the presence of 150 μm-thick cement, 37.88 MPa in 100 μm thick cement and 37.73 MPa in 50 μm-thick cement (Figure 8).

When we observe stress accumulations on the restoration surface, they were found to be 29.87 MPa in the presence of 150 μm thick cement, 36.79 MPa in 100 μm thick cement and 36.28 MPa in 50 μm thick cement (Figure 8).

In all resin cement applied groups, the least stress accumulation was found in cement material itself. It was found 1.304 MPa in the presence of 150 μm thick cement, 1.248 MPa in 100 μm thick cement and 1.696 MPa in 50 μm thick cement (Figure 8).

When the stress accumulations in the enamel tissue were compared in the groups with glass ionomer cement applied, 87.97 MPa was observed in the presence of 150 μm thick cement, 87.36 MPa in the presence of 100 μm thick cement and 89.86 MPa in the presence of 50 μm thick cement (Figure 9).

In dentin tissue after glass ionomer cement applied, 37.85 MPa stress accumulation was found in the presence of 150 μm thick cement, 37.76 MPa in the presence of 100 μm thick cement and 37.67 MPa in the presence of 50 μm thick cement (Figure 10).

After comparison of stress accumulation on the restoration surface in the groups with glass ionomer applied, 30.14 MPa was observed in the presence of 150 μm thick cement, 37.03 MPa in the presence of 100 μm-thick cement and 36.4 MPa in the presence of 50 μm thick cement (Figure 10).

In all glass ionomer-applied groups, the least stress accumulation was found in cement material itself. 1.821 MPa was observed in the presence of 150 μm-thick cement, 1.669 MPa in the presence of 100 μm thick cement and 1.981 MPa in the presence of 50 μm thick cement (Figure 10).

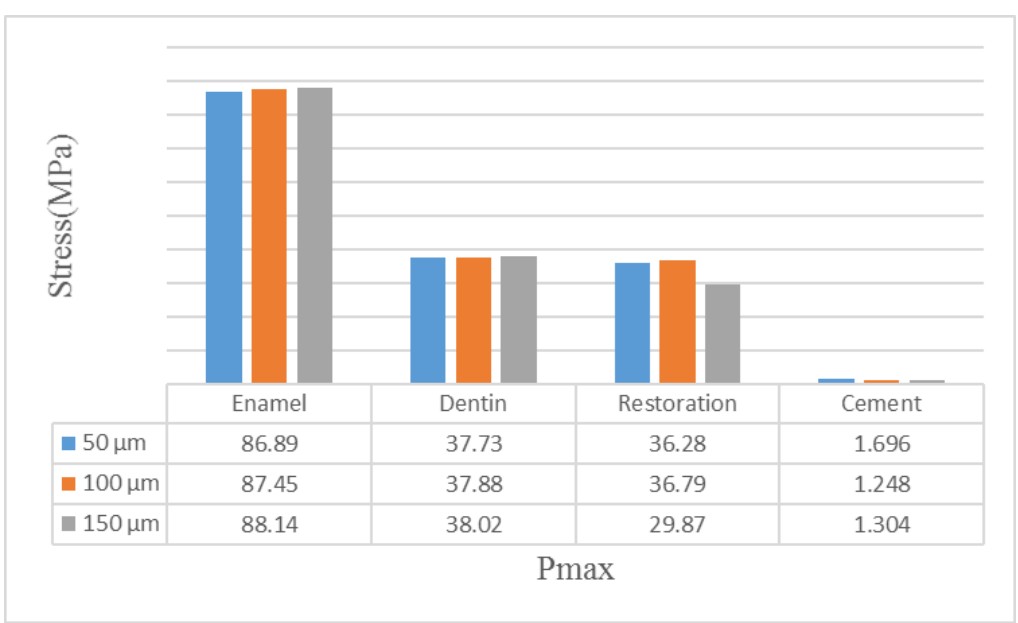

**Figure 8.** Maximum principal stress distribution in restorations applied with RC of different thicknesses (50 GPa Young's modulus of amalgam).

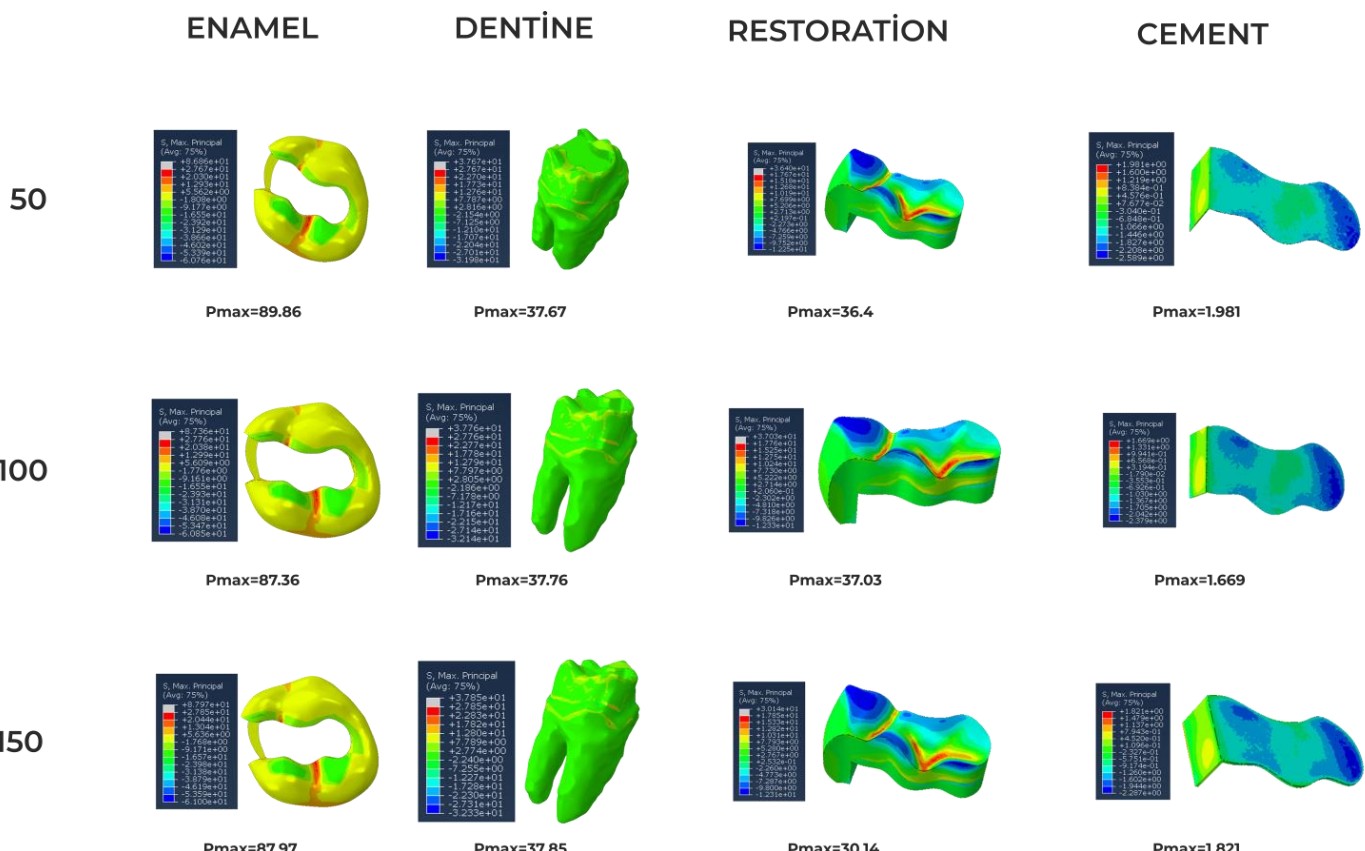

**Figure 9.** Maximum principal stress distribution in restorations applied with GI of different thicknesses (50 GPa Young's modulus of amalgam).

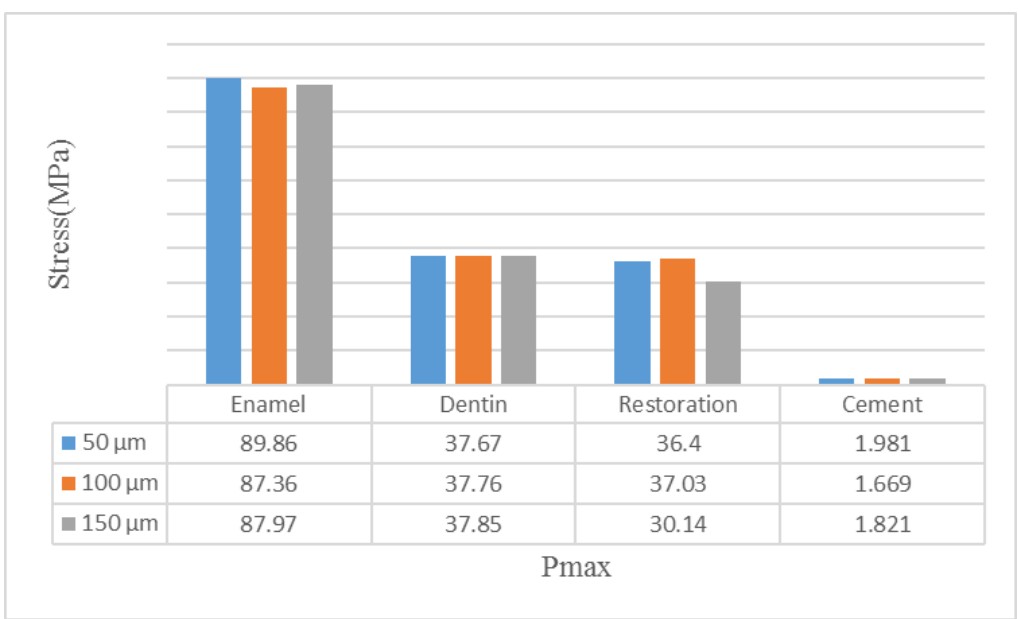

**Figure 10.** Maximum principal stress distribution in restorations applied with GI of different thicknesses (50 GPa Young's modulus of amalgam).

When the Young's modulus of amalgam was changed and also cement type and thickness were changed, it was found that in all conditions, enamel has the greatest stress accumulation and cement material has the least stress accumulation (Figure 11).

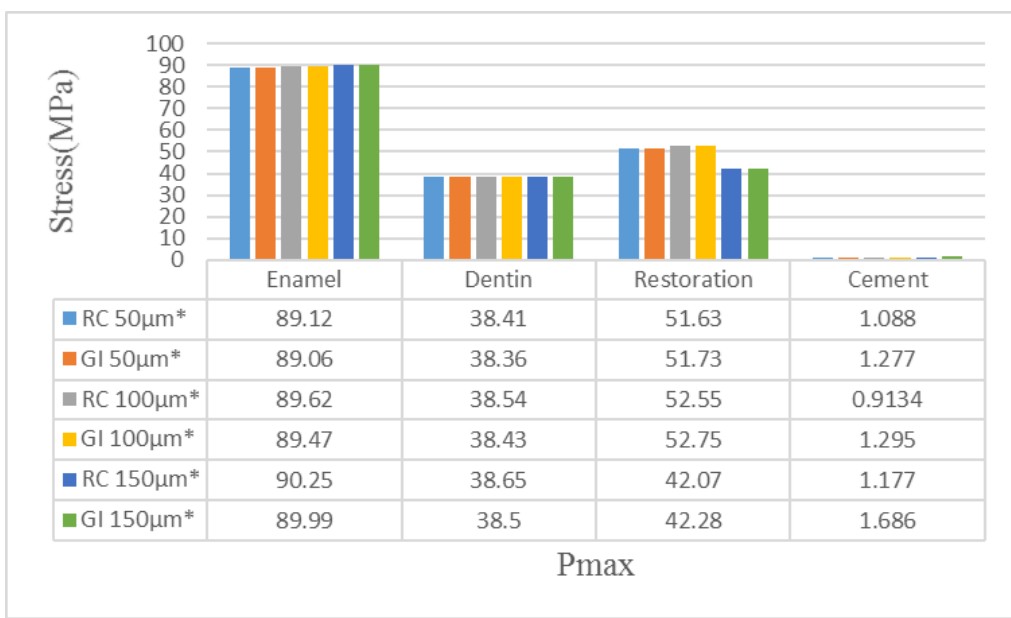

**Figure 11.** Maximum principal stress distribution in restorations with cements of different thickness (* 35 GPa Young's modulus of amalgam).

When glass ionomer cement and resin cement are compared with each other with different Young's modulus values of amalgam, there is no significant difference between the stress accumulations in enamel and dentin tissues. The least stress accumulation in enamel tissue was found when resin cement was applied in 50 μm and amalgam was applied in 50 GPa (Figure 11).

Significant differences were found in stress accumulation on restoration surface. In both cement types, the least stress accumulation was found in the presence of 150 μm-thick cement and 50 GPa amalgam presence (Figure 11).

In all groups to which there were applied different thickness of cements and different Young's modulus values of amalgam, the least stress accumulation was found in cement material itself (Figures 11 and 12).

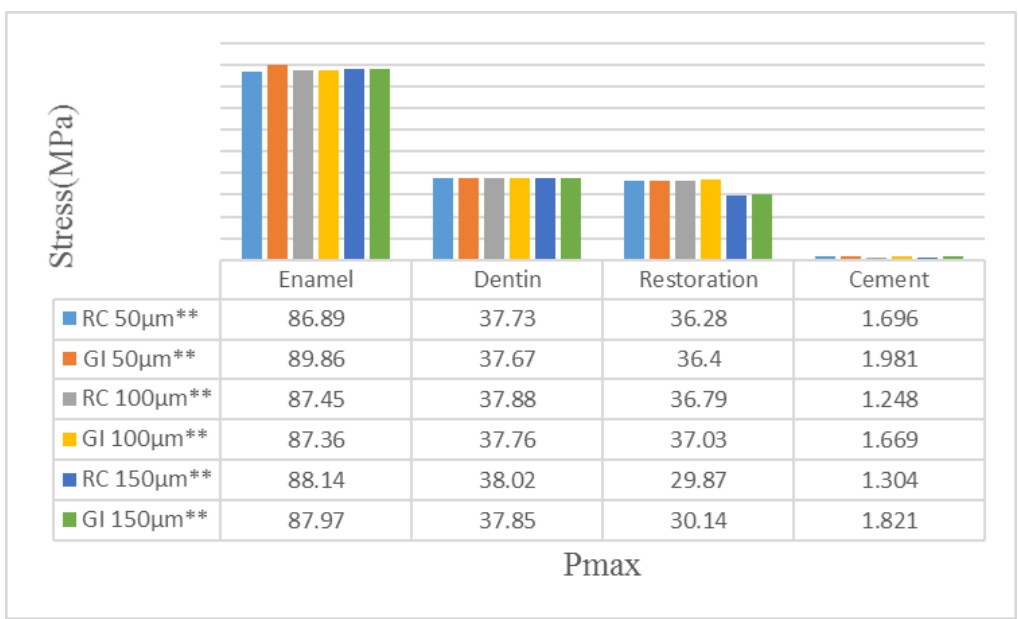

| | Enamel | Dentin | Restoration | Cement |
|---|---|---|---|---|
| RC 50μm** | 86.89 | 37.73 | 36.28 | 1.696 |
| GI 50μm** | 89.86 | 37.67 | 36.4 | 1.981 |
| RC 100μm** | 87.45 | 37.88 | 36.79 | 1.248 |
| GI 100μm** | 87.36 | 37.76 | 37.03 | 1.669 |
| RC 150μm** | 88.14 | 38.02 | 29.87 | 1.304 |
| GI 150μm** | 87.97 | 37.85 | 30.14 | 1.821 |

**Figure 12.** Maximum principal stress distribution in restorations with cements of different thickness (** 50 GPa Young's modulus of amalgam).

## 4. Discussion

Human bite forces have been investigated in several studies with various types of equipment and reported maximum values have varied widely. In a study, the maximum bite force of healthy undergraduate dental students consisting of 15 men and 15 women was investigated. Results for both sexes significantly exceeded previously reported values for one-sided housing. The mean maximum bite force value in the molar region was 847 N for men and 597 N for women. The finding that pain or lack of muscle strength often limits clenching suggests that true chewing potential has been recorded [23]. Additionally, the effect of a range of loading conditions on the stress values has been considered [24]. Generally, loads can be applied on the tooth/filling directly, in the form of concentrated [17], distributed [25], or indirectly, using a ball [26] or a morsel [27]. Even for concentrated load cases, studies showed that different parameters such as the number of loading points [28], their directions and locations [29] can result in different stress concentration values and locations [30].

Thanks to finite element analysis method, a tooth with a complicated structure can be easily examined and ideal restorations can be determined by evaluating the relationships between different materials and dental tissues using computer analysis [31]. In a study by Rodriques et al. in 2020 showed that patient specific FEA provided investigation of stress distributions in damaged tooth, allowing predictions that cannot be obtained by clinical examination [18]. Soares et al. reported that finite element analysis has provided the understanding of complicated processes and has aided researchers and clinicians in planning much better methods to enhance oral health [32]. In this study, first a molar was chosen for modelling because molars are known to constitute approximately 50% of all fractures [33].

Amalgam is still widely used in oral and dental health centers or other dental clinics in Turkey because of its ease of use and some advantages. Amalgam filling is one of the restorative materials used in posterior group teeth, where high fracture resistance and tensile properties are very important, as chewing forces are absorbed by the dental hard tissues as well as by the periodontal ligament and bone [34]. In our study, it was found that in both cement types, the least stress accumulation was found in the presence of 150 μm-thick cement and 50 GPa amalgam presence.

Dentin is more resistant to chewing forces than enamel and can absorb incoming forces better. Since enamel is a tissue with a high modulus of elasticity and low tensile strength, it is more fragile. When chewing forces are present, intact dentin may stretch a little and transfer the force (tension) to the surrounding tissues [35].

Chun et al., reported that most dental treatments aimed that restoring the functions of enamel and dentin should be carried out simultaneously and not separately. Because enamel has mechanical role such as grinding (crushing) food and its abrasion resistance is vitally important, hardness value must be prioritized before the replacement and identification of restorative materials. On the other hand, dentin absorbs bite forces which come from enamel; therefore, mechanical properties must be primarily chosen [36]. In our study, it was observed that most stress accumulation occurred in the enamel tissue.

Kim et al., used 70 μm resin cement and 100 μm resin modified glass ionomer cement [37], Ausiello et al. used 70 μm resin cement [38], He et al. used 120 μm cements [39], Neto et al. used 100 μm resin cement [40], Syed et al. used 50 μm cement [41] and Nabih et al. used 60 μm cements [42] in their studies. Based on these studies, in our studies we used 50, 100 and 150 μm thicknesses of resin and glass ionomer cements. As a result, our study showed that the presence of both glass ionomer cement and resin cement with a thickness of 150 μm caused less stress on the restoration surface.

The high tension from restorative material in the enamel of restored teeth may be due to the different modulus of the enamel and the restorative material. Similarly, it has been reported that lower stress values are obtained in teeth to which materials with the closest value to the elasticity modulus of enamel were applied [20]. In our study, when Young's modulus values of amalgam were changed and cement type and thickness were also changed, it was found that in all conditions, enamel has the greatest stress accumulation and cement material has the least stress accumulation.

In a study in which cement and restorative materials of different thicknesses were compared using finite element analysis method, it was seen that the restorative material thickness affected restoration and resin cement biomechanics. However, it was concluded that cement thickness only affects stress on the cement material itself [43]. In the results obtained in our study, it was observed that the least stress accumulation occurred in cement material by itself.

In a study evaluating the stress accumulation of different material combinations in Class 1 cavities, it was concluded that the most stress accumulation between the tooth surface and the restorative material was found in the models made with the direct technique in which GI cement and RC were used as the base material. The least stress accumulation was observed in the model using block composite [38].

Comparing the stress accumulations resulting from occlusal forces in teeth that underwent endodontic treatment using different base materials with different restorative approaches, it was found that materials with a low elastic modulus put more stress on the dental tissue [28].

## 5. Conclusions

Through modeling the first molar tooth using the finite element analysis method, stress accumulations on the restoration surface and tooth tissues were measured and evaluated. According to the results that we obtained, there was no difference between the cement types in terms of stress accumulation in the model using either glass ionomer cement or resin cement at any thickness. However, when the same cements with different thicknesses

were evaluated, it was concluded that the presence of both glass ionomer cement and resin cement with a thickness of 150 μm caused less stress on the restoration surface. On the other hand, the least stress accumulation on the restoration surface was found in the presence of 150 μm-thick cement and 50 GPa Young's modulus of amalgam.

*Clinical Implications*

Although the cement thickness cannot be precisely measured in microns in clinical situations, whenever the cement base is thicker under amalgam restorations, the better is the stress distribution. Our study showed effect of cement thickness on stress distribution in clinical use. It has been concluded that even if cement thickness measurement is not possible in restorative treatments, there should not, at least, be a very thin cement thickness. In addition, the use of amalgam with higher Young's modulus values in clinical use results in less stress distribution.

**Author Contributions:** Conceptualization, H.Y.G.; Methodology, R.M.; Software, R.M.; Formal analysis, R.M.; Data curation, M.G.D.; Writing—original draft, S.A., I.B.Y. and Y.D.F.; Writing—review & editing, H.Y.G.; Project administration, H.Y.G. All authors have read and agreed to the published version of the manuscript.

**Funding:** This research received no external funding.

**Conflicts of Interest:** The authors declare no conflict of interest.

## Abbreviations

| | |
|---|---|
| GPa | Gigapascal |
| MPa | Megapascal |
| FEA | Finite Element Analysis |
| μ | micron |
| RC | Resin cement |
| GI | Glass ionomer cement |
| N | Newton |

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
