# Peer review of "The Effects of Using Cements of Different Thicknesses and Amalgam Restorations with Different Young’s Modulus Values on Stress on Dental Tissue: An Investigation Using Finite Element Analysis"

_coatings, doi:10.3390/coatings13010006_

Round 1

Reviewer 1 Report

The work presented for review is an interesting numerical study based on an isotropic model of a tooth. Manuscript deals with comparative analysis of restorative materials for conservative dentistry. The experiment is properly designed and executed 

My few comments relate to:

In Figures 3, 5, 7, 9, the scale is poorly visible. I suggest considering enlarging the scale to readable dimensions

The discussion section is quite short. Maybe raise the issue of endocrowns in FEA studies.

Author Response

Dear reviewer, we would like to thank you for your time and review. We added Figure 3,5,7,9 as addition files after references. We raides our discussion section according to our topic.

Reviewer 2 Report

November, 25, 2022

Dear authors

Thank you for an interesting report.

In this study, you examined the stress on dental tissue in the case of using cements and amalgam restorations finite element analysis.

I agree to many parts of your claims and guessed that the subject of this paper will be of interest to the readership of Coating.  However, I think that minor revisions are required as follows:

Throughout this paper

Most of the lengths (thicknesses) used in this paper are expressed in micrometers, but throughout the paper, you sometimes use µ, sometimes µm, and sometimes having space between numerical values The expression is scattered, with or without it, and it is not unified. Conscious of reader readability, authors should unify all micrometer displays.

BACKGROUND

Since dental amalgam is not used universally and in some countries it is not used, I think that you should add not only advantages but also disadvantages. I think that you should take this into account, as this journal is an international journal and has readers not only in countries that use amalgam, but also in countries that have abolished it.

DISCUSSION and CONCLUSION

From the analysis results by the finite element method, which is the purpose of this research, the amount of stress was certainly quantified. However, it is considered that the discussion and conclusion based on results are insufficiently mentined in terms of how they contribute to clinical dentistry and how they can be used.

Figures 4, 6, 8, 10, 11

You need to correct that the unit of stress on the vertical axis is miswritten as Mpa instead of MPa..

Author Response

Throughout this paper : we corrected numerical values.

BACKGROUND: We mentioned advantages and disadvantages of dental amalgam between 103th and 116th sentences in our background section.

DISCUSSION and CONCLUSION: We mentioned about effects of our study in clinical dentistry between 352th and 355th sentences in our conclusion section

Figures 4, 6, 8, 10, 11: We corrected the MPa 

Reviewer 3 Report

Thank you for submitting your work. I find the topic outdated and of low value to the research community. Amalgam restorations should not be used after proven toxicity and development of resin modified glass ionomers with the quality of today. The research question also adds no value as the thickness of the cement below the amalgam restoration cannot be measured with microns in the clinical situation. Finally, your introduction and discussion were poor and no enough support or explanation of your results were mentioned in discussion section.

Please find the notes in the attached PDF for more comments on your work.

Author Response

Dear reviewer,we would like to thank you for your time and review. As you said, amalgam has toxicity and some disadvantages,but in many underdeveloped countries amalgam is still widely used because of economical problems. We noticed that there are no sufficient studies about this topic, for this reason we had conducted our study. In order to show the importance of using cements as base material in restorative treatments, we applied cement in different thicknesses by taking notes from the studies. We added these studies between 313th and 319th sentences in our discussion section. The notes that you added in peer review file had been noticed and corrected. Thank you for your review report.

Round 2

Reviewer 3 Report

Dear Author

Thank you for your modifications. Some comments were ignored from your side and still no sound answers were presented. Please find the attached PDF with comments. 

Regards

Author Response

Dear reviewer, thank you for your time and your review report. We raised our introduction secition as you said. We added additional informations between 83-89th and 103-109th sentences according to our topic. 

Material and methods: " This force is delivered to both restoration and tooth surface " we added this sentence according to an article which is written by Ausiello et al. ( Ausiello, P., Ciaramella, S., Martorelli, M., Lanzotti, A., Gloria, A., & Watts, D. C. (2017). CAD-FE modeling and analysis of class II restorations incorporating resin-composite, glass ionomer and glass ceramic materials. Dental Materials33(12), 1456-1465.)

Discussion: Between 313th and 319th sentences: By these articles, we aimed to determine cement thicknesses. Restoration materials that used in these articles are different than our restoration material. So, it is difficult to compare our results with other articles' results. In our study, we aimed to compare stress distribution in restoration material and tooth tissues not cement.

Conclusion: We raised and corrected our conclusion section.
